# miR766-3p and miR124-3p Dictate Drug Resistance and Clinical Outcome in HNSCC

**DOI:** 10.3390/cancers14215273

**Published:** 2022-10-27

**Authors:** Tomohiro Shibata, Duo-Yao Cao, Tahir B. Dar, Faizan Ahmed, Shabir A. Bhat, Luciana C. Veiras, Ellen A. Bernstein, Abdul Arif Khan, Manita Chaum, Stephen L. Shiao, Warren G. Tourtellotte, Jorge F. Giani, Kenneth E. Bernstein, Xiaojiang Cui, Eric Vail, Zakir Khan

**Affiliations:** 1Department of Pathology and Laboratory Medicine, Cedars-Sinai Medical Center, Los Angeles, CA 90048, USA; 2Department of Biomedical Sciences, Cedars-Sinai Medical Center, Los Angeles, CA 90048, USA; 3Department of Radiation Oncology, Cedars-Sinai Medical Center, Los Angeles, CA 90048, USA; 4Department of Pharmaceutics, College of Pharmacy, King Saud University, Riyadh 11451, Saudi Arabia; 5Department of Neurology, Cedars-Sinai Medical Center, Los Angeles, CA 90048, USA; 6Department of Surgery, Cedars-Sinai Medical Center, Los Angeles, CA 90048, USA; 7Samuel Oschin Comprehensive Cancer Institute, Cedars-Sinai Medical Center, Los Angeles, CA 90048, USA

**Keywords:** head and neck squamous cell carcinoma (HNSCC), miRNAs, miR124-3p, miR766-3p drug resistance, cisplatin, fluorouracil (5-FU), FP chemotherapy, radiotherapy

## Abstract

**Simple Summary:**

Despite improvements in therapeutics, head and neck squamous cell carcinomas (HNSCC) relapse in more than 50% of cases due to the development of chemo-radiotherapy resistance during sequential treatments. Our findings provide a strong rationale which can improve potential therapeutic strategies for overcoming drug resistance in HNSCC. We demonstrated that expression of miR124-3p and miR766-3p is associated with drug resistance in HNSCC, and their blockade greatly enhances the efficacy of standard anti-HNSCC therapeutics including 5-fluorouracil and cisplatin (FP chemotherapy), as well as radiotherapy. We discovered miR124-3p and miR766-3p-mediated mechanisms of resistance involve transcriptional factors CREBRF and NR3C2 in HNSCC. These results warrant testing miR766-3p and miR124-3p as predictors of response to chemo-radiotherapy in clinical settings and as markers for selecting patients for alternative treatment approach.

**Abstract:**

Head and neck squamous cell carcinoma (HNSCC) is a highly aggressive disease with poor prognosis, which is mainly due to drug resistance. The biology determining the response to chemo-radiotherapy in HNSCC is poorly understood. Using clinical samples, we found that miR124-3p and miR766-3p are overexpressed in chemo-radiotherapy-resistant (non-responder) HNSCC, as compared to responder tumors. Our study shows that inhibition of miR124-3p and miR766-3p enhances the sensitivity of HNSCC cell lines, CAL27 and FaDu, to 5-fluorouracil and cisplatin (FP) chemotherapy and radiotherapy. In contrast, overexpression of miR766-3p and miR124-3p confers a resistance phenotype in HNSCC cells. The upregulation of miR124-3p and miR766-3p is associated with increased HNSCC cell invasion and migration. In a xenograft mouse model, inhibition of miR124-3p and miR766-3p enhanced the efficacy of chemo-radiotherapy with reduced growth of resistant HNSCC. For the first time, we identified that miR124-3p and miR766-3p attenuate expression of CREBRF and NR3C2, respectively, in HNSCC, which promotes aggressive tumor behavior by inducing the signaling axes CREB3/ATG5 and β-catenin/c-Myc. Since miR124-3p and miR766-3p affect complementary pathways, combined inhibition of these two miRNAs shows an additive effect on sensitizing cancer cells to chemo-radiotherapy. In conclusion, our study demonstrated a novel miR124-3p- and miR766-3p-based biological mechanism governing treatment-resistant HNSCC, which can be targeted to improve clinical outcomes in HNSCC.

## 1. Introduction

Head and neck squamous cell carcinomas (HNSCC) are the sixth most common cancers worldwide. Despite advances in therapeutics, the clinical outcome of HNSCC patients has remained unchanged over the past few decades, and the overall five-year survival remains around 40–50% [1,2]. The treatment of HNSCC in advanced stages is limited to chemotherapeutic approaches and platinum-based agents, such as 5-fluorouracil (5-FU) and cisplatin (FP chemotherapy), commonly used as first- and second-line treatment options. However, the development of resistance to these anticancer drugs in tumor cells during sequential treatment is a major obstacle to their clinical application and a primary reason for the failure of head and neck cancer treatments [3,4,5,6]. For example, more than 50% of HNSCC patients respond to the chemotherapy at early stages of cancer progression, while the response rate comes down to below 30% in patients with recurrent or metastatic HNSCC [7,8]. Moreover, these drugs are highly toxic and can only be given in limited amounts. Therefore, even when being treated with first-line therapy (cisplatin + 5-fluorouracil + cetuximab), the overall survival is only ten months in patients with metastatic or recurrent HNSCC, indicating the need to explore new therapeutic regimens to treat this devastating malignancy.

A molecular understanding of how tumors develop resistance could identify novel targets to overcome resistance in HNSCC. MicroRNAs (miRNAs) are short non-coding regulatory RNAs that act as post-transcriptional repressors of gene expression through binding to the 3′-untranslated region (3′-UTR) of target mRNA in diverse biological contexts [9]. miRNAs have been found to be heavily dysregulated in many cancers, including HNSCC, and are linked to the hallmark characteristics of cancer, such as evading growth suppressors, resisting cell death, activating invasion, and metastasis [10]. Depending on their target genes, miRNAs can function as oncogenic or tumor-suppressors [11,12,13]. A growing number of studies have identified miRNAs as potential biomarkers for early cancer diagnosis and prognosis [14]. Circulating miRNAs have been implicated as promising biomarkers for the noninvasive diagnosis of various cancers [11,14,15,16]. Several miRNA-based therapeutics are currently being tested in preclinical and clinical trials [14]. For example, preclinical and clinical studies show a tumor suppressive effect of miR-16 overexpression on malignant pleural mesothelioma (MPM) [17,18]. Further, the administration of miR-16 mimics using a bacteria-derived delivery system achieved the potent inhibition of tumor growth in MPM and NSCLC patients [18]. These studies indicate that miRNAs are one of the most promising diagnostic and therapeutic targets for cancer.

This study examines the role of miRNAs in the development of drug resistance in HNSCC. By comparing responder vs. non-responder HNSCC patients, differentially expressed miRNAs were uncovered in HNSCC tumors. For the first time, we found that miR124-3p and miR766-3p are overexpressed in resistant HNSCC and are correlated with poor prognosis. Our study demonstrates that miR124-3p and miR766-3p are excellent targets for distinguishing between sensitive and resistant HNSCC and could potentially be used in determining whether patients will respond to conventional drugs or require an alternative treatment approach. Combined targeting of miR124-3p and miR766-3p has great therapeutic potential to treat resistant HNSCC and enhance the effectiveness of conventional drugs.

## 2. Materials and Methods

### 2.1. Clinical Samples

HNSCC tissues from twelve patients before adjuvant therapy were collected from Cedars-Sinai Biobank after obtaining due consent from the patients under an approved Institutional Research Ethics Board (IRB) protocol (STUDY00000244). All patients underwent adjuvant platinum-based chemo-radiotherapy treatment at Cedars-Sinai Medical between 2014 to 2016 and were followed up until death or for >4 years. Tumor size and clinical stage of HNSCC were classified according to the general rules of head and neck cancer (TNM classification). Based on the response to conventional cancer therapies and relapse, patients were classified into two groups: responder (those who were alive > 3 years after diagnosis) or non-responder (those who expired). The median overall survival (OS) for the responder cohort was 76.5 months as compared to 16.5 months for the non-responder cohort. The average OS for the responder cohort was 96.8 months as compared to 16.7 months for the non-responder cohort. The clinical characteristics of these patients are shown in Table 1. At the initial diagnosis, the clinical stage ranges from T2 to T4 and T1 to T4a in responder and non-responder groups, respectively. For both groups, nodal status ranges from N0–N2b and no metastasis was identified (Table 1). The initial procedure of all samples included biopsy (10/16), local resection (5/16), and neck dissection (1/16). Two patients in the responder group had metastatic disease rendered from re-resection but without lymph node metastasis. Normal, uninvolved tissue from the same site was collected and used as control.

### 2.2. RNA Purification from FFPE Samples

A representative section of every sample was stained with H&E and reviewed by a pathologist to identify regions containing > 70% malignant epithelial cells for macrodissection. Adjacent normal tissue from patients was dissected from the proximal tumor resection margin at a distance of around 1 cm. Total RNA enriched for small RNA species was isolated using the miRNeasy FFPE Kit (Qiagen) according to the manufacturer’s instructions. All blocks were processed randomly, with clinical outcomes unknown, to avoid experimental bias.

### 2.3. miRNA Sequencing

After confirming RNA quality (RNA integrity number (RIN) > 7) using 2100 Bioanalyzer, samples were processed for global human miRNA microarray profiling. This analysis provides expression data of all known human miRNAs and their isotypes (~2500). RNA integrity was analyzed on the 2100 Bioanalyzer using the Agilent RNA 6000 Pico or Nano Kit (Agilent Technologies, Santa Clara, CA, USA), and RNA was quantified using the Qubit RNA HS Assay Kit (ThermoFisher Scientific, Waltham, MA, USA). The QIASeq^TM^ miRNA Library Kit (Qiagen, Hilden, Germany) was used to prepare miRNA sequencing libraries. Final library concentrations were measured using the Qubit 1X dsDNA HS Assay kit. Library fragment size was determined on the 2100 Bioanalyzer or Agilent 4200 TapeStation (Agilent Technologies), and then libraries were sequenced on a NovaSeq 6000 (Illumina, San Diego, CA, USA) at an average sequencing depth of ~10 M reads/sample and 1 × 75 bp sequencing.

### 2.4. Data Analysis

The demultiplexed raw reads were uploaded to GeneGlobe Data Analysis Center (Qiagen, at https://www.qiagen.com/us/resources/geneglobe; accessed on 22 October 2020) for quality control, alignment, and expression quantification. Briefly, 3′ adapter and low-quality bases are trimmed off from reads first using cutadapt (version 1.13) with default settings, then reads with less than 16 bp insert sequences or with less than 10 bp UMI sequences are discarded. The remaining reads were collapsed to UMI counts and aligned to miRBase (release v21) mature and hairpin databases sequentially using Bowtie v1.2 [19,20]. The UMI counts of each miRNA molecule are counted, and the expression of miRNAs is normalized based on the total UMI counts for each sample. Differentially expressed miRNAs were compared between all three groups and presented as Venn diagram and heat map using GraphPad Prism 7. The miRNA expression profile is listed in Appendix A.

### 2.5. Gene Ontology (GO) Analysis of miRNA Target Genes

Functional enrichment for biological processes within the GO pathway was calculated via DAVID database (https://david.ncifcrf.gov/; accessed on 30 April 2021), which is an online tool for gene annotation, function visualization, and large-volume data integration [21,22]. We identified miRNA direct target genes which have a target score of more than 95 using miRDB and Ingenuity Pathway Analysis (IPA). All miRNA targets have a prediction score in the range of 50–100 as assigned by miRDB and more than 95 higher scores representing statistical confidence in the prediction result [23]. All miRNA target genes using GO analysis are listed in Appendix A. GO clusters describe gene products with three independent categories (Biological process, Molecular Function, and Cellular Component) and describe gene product attributes.

### 2.6. The Cancer Genome Atlas (TCGA) Dataset

The TCGA mRNA expression datasets of HNSCC cancer patients (n = 488) were obtained from two TCGA datasets (TCGA—PanCancer Atlas; TCGA—Firehose Legacy) (the cBioPortal for Cancer Genomics, http://www.cbioportal.org; accessed on 28 April 2021) [24,25]. To determine the relationship between miRNA-target gene expression, drug resistance, and overall survival, HNSCC patients in each cohort were divided into two mRNA low- or high-expression groups. We performed Kaplan–Meier survival analysis using GraphPad Prism 7 software. The clinical background and expression of mRNA are listed in Appendix A.

### 2.7. Kaplan–Meier Plotter Dataset

The prognostic significance of mRNA expression in HNSCC (n = 499) was evaluated by using Kaplan–Meier plotter (http://kmplot.com/analysis/; accessed on 30 April 2021) [26]. For this analysis, system acquired mRNA array databases from GEO and EGA. Clinical samples are divided into low and high miRNA expression groups according to median values of mRNA expression and assessed by Kaplan–Meier survival plot.

### 2.8. Cell Culture and Transfection

HNSCC cell lines CAL27 and FaDu were obtained from ATCC. These cell lines were cultured at 37 °C in Dulbecco’s modified Eagle’s medium (DMEM) supplemented with 10% fetal bovine serum (FBS) in a humidified atmosphere containing 5% CO_2_. To investigate the effects of selected miRNAs, inhibitors and mimics were purchased from Qiagen as follows: Negative Control miRCURY LNA miRNA Mimic (YM00479902-ADA), hsa-miR-124-3p miRCURY LNA miRNA Mimic (YM00471256-ADA), hsa-miR-766-3p miRCURY LNA miRNA Mimic (YM00472972-ADA), mirVana™ miRNA Inhibitor Negative Control (4464076), hsa-miR-124-3p inhibitor (MH10060), and hsa-miR-766-3p inhibitor (AM11856). To investigate the effects of miRNA target genes, siRNAs were purchased from ThermoFisher Scientific as follows: siRNA negative control (12935200), NR3C2 siRNA (s533953), and CREBRF siRNA (s45763). Cancer cell lines were transfected using Lipofectamine RNAiMAX (ThermoFisher Scientific, Waltham, MA, USA) according to the protocol provided by the manufacturer.

### 2.9. Establishment of Resistant Cell Lines

CAL27 resistant subline was selected based on constant exposure of the parental cells to the single drug or combination of cisplatin (Millipore Sigma) and 5-FU (Millipore Sigma) in a stepwise dose incremental strategy. The half maximal inhibitory concentration (IC_50_) of each drug was calculated by MTT assay. CAL27 cells were treated with a sequential increase in dosage of the two drugs ranging from 1 μM to 6 μM. Resistant cell line was established from different flasks and was not cloned.

### 2.10. Quantitative Real-Time PCR

The miRNA was extracted from cancer cell lines using miRNeasy Kit (Qiagen), and then 10 ng RNA was used from each sample for cDNA synthesis using miRCURY LNA RT Kit (Qiagen). Quantitative real-time PCR of the individual miRNA was performed in a total volume of 10 μL containing 5 μL 2X miRCURY SYBR Green MasterMix (Qiagen), 0.05 μL ROX Reference Dye, 1 μL Primer mix, 1 μL RNase free water, and 3 μL cDNA (60 times diluted in nuclease-free water). Amplification was performed in quadruple in Quantstudio5 (ABI) using PCR program as follows: 2 min at 95 °C, and then 40 cycles of 10 s at 95 °C and 60 s at 56 °C. Specific primers were purchased from Qiagen (hsa-miR-124-3p # YP00206026, hsa-miR-766-3p # YP00204499). The miRNA levels were normalized to the stable internal control small nuclear RNA RNU5G (YP00203908).

### 2.11. Colony Formation Assay

Cells transfected with miRNA inhibitor (10 nM) or miRNA mimic (10 nM) for 24 h were seeded in 24-well plates (1.5 × 10^3^ cells), irradiated at 0 or 8 Gy (Mueller RT-250 γ-ray, Thoraeus Filter, 200 kV, 10 mA), and grown in normal media with or without cisplatin or 5-FU for 7 days, washed once with 1X PBS, fixed with methanol, and stained with 0.5% crystal violet.

### 2.12. MTT Assay

Cell viability was evaluated using 3-[4,5-dimethylthiazol 2-yl]-3,5-diphenyl tetrazolium bromide (MTT) test (R&D Systems). Exponentially growing cell suspensions were seeded into 96-well plate (1.5 × 10^3^ cells/well) and the next day drug was added at indicated concentration. After incubation for 72 h, 10 μL of MTT reagent was added to each well, and the plates were incubated for 4 h at 37 °C. Then, 100 μL of detergent reagent was added, and post 2 h incubation at 37 °C, cytotoxicity was calculated by recording absorbance at 570 nm with microplate reader (Omega). Each value represents the average from triplicate wells ± SD. The IC_50_ value for each drug was calculated from concentration–response curves using the non-linear regression mathematical equation.

### 2.13. Wound Healing Migration Assay

Cells were seeded in a 24-well plate (5.0 × 10^4^ cells cells/well) and incubated in 10% FBS DMEM. At confluence, the cell monolayer was scratched using 20 mL sterile pipette tip. We then washed away detached cells with PBS and added DMEM without FBS. Images were captured using an inverted microscope (magnification, ×40) post 0 and 24 h incubation. The width of the wound was then measured using ImageJ software as described previously [27]. The equation for calculation of relative wound area is following: Relative wound closure (%) = ((W_0_ − W_t_)/W_0_) × 100; W_0_ = Wound are at 0 h (mm^2^) W_t_ = Wound are at t h (mm^2^).

### 2.14. Transwell Migration and Invasion Assays

Cells were seeded in the upper chamber of transwell inserts (8 µM pore size; Corning) at indicated number in 500 µL serum-free media, and then the inserts were put onto 24-well plate containing 700 µL media with 10% serum. The plates were then incubated for 24 h at 37 °C. For counting cells in the lower chamber, remaining media and cells that had not migrated from the top of the membrane were removed with a cotton-tipped applicator. Then, migrated cells on the undersurface of the inserts were fixed in methanol and stained with crystal violet for cell counting.

For invasion assay, transwell inserts were coated with 0.05% Matrigel (BD Biosciences, NJ, USA) and incubated at room temperature for 4 h before seeding the cells in upper chamber. Rest all conditions were same as described for migration assay.

### 2.15. Measurement of Apoptosis

Cells were harvested and stained with Annexin V and propidium iodide (PI) (BioVision, MA, USA) for 5 min at room temperature and then analyzed with flow cytometry (CYTEK NL-3000) to determine cellular apoptosis. Experiments were performed in triplicate, and statistical analysis was performed.

### 2.16. Western Blotting Analyses

Cells were washed twice with cold PBS and lysed with RIPA buffer containing protease and phosphatase inhibitors (ThermoFisher Scientific). The polyvinylidene difluoride membranes were incubated with specific antibodies, including CREBRF rabbit monoclonal antibody (mAb) (ThermoFisher, PA5-68552; 1:1000), CREB3 rabbit mAb (Proteintech, 11275-1-AP; 1:1000), NR3C2 mouse mAb (Novus, NB300-562; 1:1000), c-Myc rabbit mAb (Novus, NB600-336; 1:1000), ATG5 rabbit mAb (Novus, NB110-53818SS; 1:1000), β-catenin mouse mAb (Novus, NBP1-54467SS; 1:1000), and β-actin mouse mAb (Sigma-Aldrich, A3854; 1:1000). The membranes were incubated with IRDye^®^ 680RD Goat anti-Rabbit IgG Antibody (LI-COR Biosciences, 926-68071; 1:5000) or IRDye^®^ 800CW Goat anti-Mouse IgG Antibody (LI-COR Biosciences, 926-32210; 1:5000). Protein bands were measured using an Odyssey Infrared Imaging System (ODYSSEY CLx, Li-COR). The fluorescence intensity was evaluated using Image Studio Lite version 5.2.

### 2.17. Immunohistochemical Analysis

Paraffin-embedded tissue blocks were produced in accordance with optimal staining standards at the immunohistochemistry lab at Cedars-Sinai Medical Center. To investigate molecular pathways relevant to miR124-3p and miR766-3p expression, the histology sections (4 μm) were obtained from responder and non-responder tumors. The slides were heated for antigen retrieval in 10 mM sodium citrate (pH 6.0). Sections were subsequently exposed to specific antibodies for CREBRF rabbit mAb (ThermoFisher, PA5-68552; 1:100), CREB3 rabbit mAb (Proteintech, 11275-1-AP; 1:100), ATG5 rabbit mAb (Novus, NB110-53818SS; 1:100), NR3C2 mouse mAb (Novus, NB300-562; 1:100), β-catenin mouse mAb (Novus, NBP1-54467SS, 1:100), and c-Myc rabbit mAb (Novus, NB600-336; 1:100). The slides were exposed to biotinylated goat anti-rabbit or anti-mouse IgG (VECTASTAIN, PK-6101-NB; 1:500), and then detected them using Elite^®^ ABC-HRP Kit (PK-6101-NB) and DAB Substrate Kit (VECTASTAIN, SK-4100). All slides were subsequently counterstained with Mayer’s hematoxylin. Determination of CREBRF, CREB3, ATG5, NR3C2, β-catenin, and c-Myc staining areas were described previously [28].

### 2.18. Mice

All animal experimental protocols were approved by the Cedars-Sinai Institutional Animal Care and Usage Committee (IACUC010170). Male BALB/c nu/nu athymic nude mice (aged 6–7 weeks) were purchased from Jackson Lab (Strain 002019) and housed in microisolator cages under a 12-hour light/dark cycle. Water and food were supplied ad libitum. Animals were observed for tumor growth, activity, feeding, and pain according to the guidelines of the Harvard Medical Area Standing Committee on Animals.

### 2.19. Xenograft Study

CAL27/FP-R cells (5.0 × 10^6^ cells/injection) were subcutaneously injected into both flanks of 6 weeks old male nude mice groups. Tumor sizes were measured, and tumor volumes (mm^3^) were calculated as follows: length × width^2^ × 0.5. When tumors reached 50 mm^3^, mice were randomly allocated into groups (n = 6–8 per group). Vehicle, miR124-3p inhibitor, miR766-3p inhibitor, and cisplatin (2.5 mg/kg), alone or combined, were injected. Cisplatin was injected i.p. into mice on every 3^rd^ day, starting from day 7 post-xenograft implantation to day 33. In vivo-jetPEI (Polyplus Transfection, Illkirch, France) was used as a delivery agent to administer miRNA inhibitors (1 µg) intratumorally at every 3rd day (miRNA 10 μg and in vivo-jetPEI reagent 1.2 µL in 100 µL of 5% glucose for one injection). The tumors were harvested after 33 days, stored at −80 °C, or immediately fixed in 10% paraformaldehyde overnight at 4 °C.

### 2.20. Statistical Analysis

Experimental results were expressed as mean ± SD or SEM. Statistical differences between groups were assessed by two-tailed Student t test. In comparing multiple groups, one-way or two-way ANOVA with Bonferroni’s correction was applied for all pairwise comparisons. Overall survival rates were analyzed using Kaplan–Meier survival analysis, and the statistical significance of between-group differences was evaluated using the log-rank test. The criteria of *p*-value < 0.05 was determined to be significantly different. The statistical analyses were conducted by GraphPad Prism 7 software.

## 3. Results

### 3.1. Overexpression of miR124-3p and miR766-3p Is Associated with Poor Prognosis and Drug Resistance in HNSCC Patients

To identify the key miRNAs related to HNSCC chemo-radiation therapy resistance, we compared miRNA profiling between responder (sensitive to chemo-radiotherapy, n = 6) and non-responder (resistant to chemo-radiotherapy, n = 6) patients. The differentially expressed miRNAs were sorted with the criteria for both *p* value < 0.05 and fold change (FC) > 2.0. We identified differential expression of 105 miRNAs between all three comparisons (responder vs. normal, non-responder vs. normal, and non-responder vs. responder) (Figure 1A). Among them, 81 miRNAs were overexpressed in tumors as compared to normal tissues, and 22 miRNAs overexpressed in non-responder (resistant) HNSCC in comparison to both responder (sensitive) HNSCC and normal tissues as shown by the Venn diagram (Figure 1B). Visualization of RNA-Seq results with Volcano Plot showed that 24 miRNAs were significantly differentially expressed between non-responder vs. responder HNSCC tumors, and all of them were increased in resistant tumors as compared to the sensitive tumors (Figure 1C).

For further analysis, we selected the top four miRNAs (miR124-3p, miR766-3p, miR6787-5p, and miR3168) that are significantly upregulated (*p* < 0.01) in non-responder HNSCC as compared to responder tumors (Figure 1D). To understand if these miRNAs regulate cancer and drug resistance pathways, we performed a miRNA direct target sequence analysis using miRbase to identify the predictive target genes (Appendix A) [21], which then was analyzed by Gene Ontology (GO) to identify their association with cancer progression. We found that miR766-3p and miR124-3p targeted genes are significantly associated with gene clusters of DNA repair, cell migration, and transcription, which are associated with tumor aggressiveness, including response to DNA-damaging agents (Figure 1E). However, miR6787-5p and miR3168 target genes are not significantly related to a GO cluster of tumor progression (Appendix A). These data indicate that miR124-3p and miR766-3p could induce resistance to DNA-damaging anticancer drugs such as 5-FU and cisplatin through the regulation of the above gene clusters.

### 3.2. miR124-3p and miR766-3p Confer Resistance of HNSCC Cells to Cisplatin and 5-FU

Next, we examined the effects of miR124-3p and miR766-3p on the sensitivity of HNSCC cells to chemotherapy. CAL27 and FaDu HNSCC cell lines were treated with either cisplatin or 5-FU, standard chemotherapeutics for HNSCC. The IC_50_ for cisplatin and 5-FU was 5 µM and 2.1 µM for CAL27 cells and 15.7 µM and 26.1 µM for FaDu cells, respectively (Figure 2A). Since the FaDu cell line is less sensitive to these drugs with increased IC_50_ as compared to the CAL27 cell line, the levels of miR124-3p and miR766-3p were also significantly higher in FaDu cells as compared to CAL27 cells (Figure 2B), indicating that miR124-3p and miR766-3p expression levels were correlated with cell sensitivity to cisplatin and 5-FU. To investigate the role of miR124-3p and miR766-3p on drug resistance, HNSCC cell lines were transfected with either miRNA inhibitors or mimics. As shown in Figure 2C, the inhibitors efficiently reduced the expression, while the mimics overexpressed each targeted miRNA. The inhibition of miR124-3p or miR766-3p increased the anti-cancer effects of both cisplatin and 5-FU on CAL27 and FaDu cells. The IC_50_ concentrations of cisplatin and 5-FU were reduced by 2.7–5.3 fold with each inhibitor in both cell lines. We also noticed that the inhibitor effects were additive, meaning that when cancer cells were pretreated with a combination of miR124-3p and miR766-3p inhibitors, the sensitivity of cancer cells to cisplatin or 5-FU was almost doubled as compared to single inhibitor treatment (Figure 2D and Appendix A). In contrast to inhibition, overexpression of miR124-3p and miR766-3p following mimics treatment provided strong resistance to cisplatin and 5-FU mediated cytotoxicity of CAL27 and FaDu cells (Figure 2E and Appendix A).

We also examined the effect of miR124-3p and miR766-3p expression on apoptosis. Cisplatin or 5-FU treatment induced apoptosis in HNSCC cells. Inhibition of miR124-3p or miR766-3p by inhibitors significantly increased apoptosis in CAL27 and FaDu cells following cisplatin or 5-FU treatment (Figure 3A and Appendix A, *p* < 0.01). For cisplatin, but not for 5-FU, these effects were additive as the apoptotic rate was doubled on treating the cells with both the inhibitors in combination. In contrast, miR124-3p and miR766-3p mimic treatment alone or in combination with significantly suppressed drug-induced apoptosis of HNSCC cells (Figure 3B and Appendix A).

Finally, we tested the effect of miRNA expression on the clonogenic potential of HNSCC cells. Results from clonogenic assays show that inhibition of miR124-3p or miR766-3p reduced the number of colonies formed by CAL27 and FaDu cells after treatment with cisplatin or 5-FU, while overexpression of miR124-3p or miR766-3p increased the number of colonies (Figure 3C,D).

### 3.3. miR766-3p Increases HNSCC Cell Migration and Invasion

Next, we examined whether miR124-3p and miR766-3p affect cell migration and invasion of HNSCC cells. In wound healing migration assays, the miR766-3p inhibitor significantly suppressed migration of CAL27 and FaDu cells by 2–3 fold as compared to the control group (*p* < 0.01), but the miR124-3p inhibitor did not affect cell migration (Figure 4A). In contrast, the miR766-3p mimic significantly increased cell migration (*p* < 0.01). Consistently, a transwell migration assay also demonstrated a positive correlation between miR766-3p expression and cell migration. miR766-3p inhibition was suppressed, while mimic overexpression increased cell migration of CAL27 and FaDu cell lines (Figure 4B). We further assessed the effect of the miRNA inhibitor or mimic on cell invasion using a Matrigel-coated transwell chamber. As shown in Figure 4C, inhibition of miR766-3p decreased invasion of CAL27 and FaDu cells by 4–7 fold (*p* < 0.01), while the miR766-3p mimic significantly increased invasion of HNSCC cells (Figure 4C, *p* < 0.01). Again, inhibition or overexpression of miR124-3p had no significant effect on HNSCC cell invasion (Figure 4C).

### 3.4. miR124-3p and miR766-3p Support Acquired Resistance in HNSCC Cells

To reveal whether miR124-3p and miR766-3p expression is associated with acquired resistance, we established a multidrug-resistant cell line (CAL27/FP-R) for cisplatin and 5-FU (Figure 5A,B). The IC_50_ dose for CAL27/FP-R cell line increased by 10–12-fold for cisplatin (34.6 µM) and 5-FU (23.7 µM), respectively, as compared to their parental cell line CAL27 (3.6 µM and 1.9 µM for cisplatin and 5-FU, respectively) (Figure 5A). CAL27/FP-R cells also showed strong resistance to 5-FU and cisplatin combination treatment with significantly less cytotoxicity as compared to CAL27 cells (Figure 5B). This resistant cell line shows aggressive behavior. For example, CAL27/FP-R showed increased migration as compared to the original CAL27, measured by transwell assays (Figure 5C). We observed a direct link between acquired resistance and expression of miR124-3p and miR766-3p. The levels of both miRNAs significantly increased in the resistant CAL27/FP- R cell line as compared to CAL27 cells following the acquisition of multidrug resistance (Figure 5D).

To validate whether miR124-3p and miR766-3p targeting can be used for sensitizing resistant cancer cells to chemotherapy, we determined the effect of miRNA inhibitors on CAL27/FP-R response to FP therapy. Cells were treated with miRNA inhibitors for 24 h before challenge with drugs, which completely blocked the expression of miR124-3p and miR766-3p (Appendix A). We found that inhibition of miR124-3p and miR766-3p significantly overcame the resistance phenotype of the CAL27/FP-R cell line with increased response to cisplatin and 5-FU treatment (Figure 5E and Appendix A) and decreased cell migration (Figure 5F,G).

Because the standard approach to HNSCC treatment includes radiotherapy, we evaluated the effect of miR124-3p and miR766-3p inhibition on chemotherapy and radiotherapy alone as well as in combination. After transfection with miR124-3p and/or miR766-3p, CAL27/FP-R cells were treated with either radiotherapy alone (8-Gy) or in combination with chemotherapy (3 µM cisplatin or 5-FU). We found a 2.4-fold increase in apoptosis when cells were transfected with combined miR124-3p and miR766-3p inhibitors in response to radiation treatment (Figure 6A and Appendix A). Similarly, the apoptotic rate was significantly higher in combination treatment as compared to single therapy (radiation or chemotherapy), which was further increased ~3-fold with the blockade of miR124-3p and miR766-3p. In contrast, miRNA inhibition significantly decreased the survival rate and clonogenic potential of CAL27/FP-R cells in response to chemo-radiotherapy treatment (Figure 6B,C). These results indicate that increased expression of miR124-3p and miR766-3p is directly associated with acquired resistance in HNSCC cells, and blockade of their expression significantly sensitizes resistant HNSCC cells to chemo-radiotherapy.

### 3.5. Inhibition of miR124-3p and miR766-3p Enhances the Effect of Chemo-Radiotherapy on Resistant HNSCC Xenograft

To validate the regulatory role of miR124-3p and miR766-3p in chemo-radiation resistance in vivo, we examined the effects of miR124-3p and miR766-3p inhibition on resistant tumor growth using a xenograft mouse model. CAL27/FP-R cells were injected subcutaneously in immunocompromised nude mice (NU/J), and tumor growth was monitored with or without cisplatin and/or radiation exposure. For miRNA inhibition, mice were injected intratumorally every three days with miR124-3p and miR766-3p inhibitors, starting from day 7 to day 33. This treatment markedly reduced the expression of miR124-3p and miR766-3p in tumors, as measured by qRT-PCR (Appendix A). Because CAL27/FP-R cells resist chemotherapy, the clinical dose of cisplatin alone showed no effect on tumor growth, while 8 Gy radiation exposure with or without cisplatin reduced tumor volume and weight by 20–30%, as compared to the naïve group (Figure 7A–C). In contrast, inhibition of miR124-3p and miR766-3p significantly enhanced the anti-cancer effect of cisplatin or radiation with reduced tumor volume (87.2 mm^3^ and 108 mm^3^, respectively) and tumor weight (79 mg and 106 mg, respectively) as compared to the naïve group (volume: 416 mm^3^ and weight: 305 mg) at day 33 post-tumor implantation (Figure 7A–C and Appendix A). Further, the effect of combination therapy, including miRNA inhibitors and chemo-radiotherapy, was examined. We found an additive anti-cancer interaction between miRNA inhibitors, cisplatin, and radiation leading to an enormous reduction in tumor growth (volume: 32.1 mm^3^ and weight 42 mg). There was no marked weight loss in single or combination therapy-treated mice (Figure 7D). These in vivo data indicate that combined targeting of miR124-3p and miR766-3p has immense potential to overcome drug resistance in HNSCC.

### 3.6. miR124-3p and miR766-3p induce Drug Resistance and Poor Prognosis in HNSCC via CREBRF and NR3C2 Mediated Pathways

To determine the mechanism of how miR124-3p and miR766-3p regulate the HNSCC response to cancer therapies, the functional analyses of miRNA target genes were performed with GO enrichment analysis. All GO term genes for miR124-3p and miR766-3p mediated pathways are presented in Appendix A. Then, by analyzing the TCGA database and Kaplan–Meier plotter (KM plotter) database, all GO term genes were correlated with HNSCC patients’ survival. Because resistant HNSCC tumors overexpress miR124-3p and miR766-3p, we expected a down-regulation of their direct target genes. Out of all GO term genes, we found that low expression of CREBRF and NR3C2 was significantly associated with poor survival of HNSCC patients (Appendix A–F). However, the other GO term genes were not significantly correlated with patients’ survival. Moreover, CREBRF and NR3C2 are included in the GO terms of DNA repair and transcription, which are mechanisms regulating cellular growth and response to DNA-damaging agents (Appendix A). Therefore, we predicted that overexpression of miR124-3p and miR766-3p induce treatment resistance in HNSCC by suppressing expression of CREBRF and NR3C2, respectively. Indeed, three-prime untranslated region (3′UTR) sequence analysis using the miRBase database predicted that miR124-3p and miR766-3p can bind to the 3′UTR of CREBRF and NR3C2 mRNA, respectively, suggesting that CREBRF and NR3C2 are direct targets of miR124-3p and miR766-3p (Appendix A). Next, we analyzed the TCGA database to examine the relation between miRNA expression and mRNA levels of their target genes in patients. As predicted, we found that expression of miR124-3p and miR766-3p is inversely correlated with the mRNA levels of CREBRF and NR3C2, respectively, in patients HNSCC (Appendix A).

CREBRF and NR3C2 transcriptional factors are considered tumor suppressors [29,30,31,32,33]. While CREBRF suppresses tumor progression by downregulating CREB3 and ATG5, NR3C2 inhibits cancer cell survival and migration through repression of β-catenin and c-Myc [29,30,31,32,33]. Our analysis from both the TCGA database and KM plotter database indicated that upregulation of CREB3/ATG5 and β-catenin/c-Myc pathways are associated with poor survival of HNSCC patients (Appendix A). To validate whether miR124-3p and miR766-3p enhance the resistant phenotype via the CREBRF-CREB3/ATG5 and NR3C2-β-catenin/c-Myc pathways, we examined the expression of these signals in HNSCC in relation to therapeutic response. While miRNA inhibitors induced, mimic treatment reduced expression of CREBRF and NR3C2 in CAL27 and FaDu cells, measured by Western blot analysis (Figure 8A and Appendix A). As expected, the expression of downstream targets of CREBRF (CREB3 and ATG5) and NR3C2 (β-catenin and c-Myc) was high with respect to the inhibition of miR124-3p and miR766-3p. In contrast, mimic treatment increased the expression of CREB3/ATG5 and β-catenin/c-Myc in HNSCC cells. Because the resistant HNSCC cell line CAL27/FP-R overexpress miR124-3p and miR766-3p, these cells showed reduced levels of CREBRF and NR3C2 and increased levels of CREB3/ATG5 and β-catenin/c-Myc as compared to control CAL27 cells (Figure 8B and Appendix A). To confirm the role of CREBRF and NR3C2 in HNSCC drug resistance, FaDu and CAL27 cell lines were treated with either CREBRF and NR3C2 siRNA alone or in combination, and the sensitivity of these cells to cisplatin and 5-FU was determined. The CAL27/FP-R resistant cell line was used as a positive control for drug response. The gene silencing was confirmed by Western blot, which showed approximately 80–90% reduction of CREBRF and NR3C2 expression following 24 h treatment with the respective siRNA (Appendix A). We found that CREBRF and NR3C2 knockdown converts relatively drug-sensitive CAL27 and FaDu cells into drug-resistant cells (equivalent to resistant CAL27/FP-R cells). Following treatment with CREBRF-siRNA or NR3C2-siRNA, the IC_50_ for cisplatin and 5-FU was increased by 2–4-fold, which was further increased (6–13-fold) with the combined inhibition of CREBRF and NR3C2 (Figure 8C and Appendix A). Similarly, the rate of drug-induced apoptosis was significantly decreased with CREBRF and/or NR3C2 knockdown (Figure 8D, *p* < 0.05). However, the suppression of apoptosis was higher with NR3C2 inhibition as compared to CREBRF inhibition. These results demonstrated a direct link between CREBRF and NR3C2 expression and HNSCC drug resistance.

Finally, we examined expression of CREBRF-CREB3/ATG5 and NR3C2-β-catenin/c-Myc signals in responder vs. non-responder HNSCC patients. Histological analysis showed that expression of CREBRF and NR3C2 was significantly reduced in non-responder HNSCCs as compared to responder HNSCCs (Figure 9A). While in relation to low CREBRF and NR3C2, the expression of downstream targets—CREB3/ATG5 and β-catenin/c-Myc—was significantly higher in non-responder HNSCCs. To examine tumor morphological changes in relation to these pathways, we performed an analysis of serial histological sections. In responders, we observed a clear border between the tumor and the surrounding normal tissue (Figure 9B). In contrast to responders, CREB3/ATG and β-catenin/c-Myc expression was high in non-responder tumor cells, which was associated with increased invasion of cancer cells into blood vessels and surrounding normal tissue (red arrow), indicating aggressive behavior of resistant cancer cells (Figure 9B). These data suggest that the expression of miR124-3p and miR766-3p and their target genes can predict poor prognosis and therapeutic response in HNSCC patients.

## 4. Discussion

Although cisplatin and 5-FU are standard therapy for HNSCC, tumor relapses in more than 50% of cases due to the development of resistance to chemo-radiotherapy [1,34,35]. Therefore, it is imperative to understand the resistant mechanisms of HNSCC to therapeutic strategies. In the past decade, miRNA, single-stranded RNA with multiple regulatory functions, emerged as a key regulator in the initiation and progression of various human malignancies, including HNSCC. Due to their ability to influence oncogene and/or tumor suppressor genes leading to the deregulation of cellular physiologic function, miRNAs play a crucial role in cancer etiology. In this study, we performed miRNA profiling of sensitive vs. resistant HNSCCs from patients and identified key miRNAs related to chemo-radiation therapy resistance. We discovered that upregulation of miR124-3p and miR766-3p is associated with poor prognosis and therapeutic resistance. This demonstrates that miR124-3p and miR766-3p targeting could be an effective tool to overcome drug resistance in HNSCC.

MiR124 has been revealed to play important roles in proliferation, apoptosis, drug resistance, and metastasis in several cancers [36]. However, contrary to our findings, the majority of studies are focused on miR124 as a tumor suppressor, and it is generally accepted that miR124 inhibits cell proliferation and metastasis of cancer through suppression of several oncogenes such as STAT3, ROCK1, Slug, and ZEB [36]. Moreover, in HNSCC, Zhao and colleagues reported that miR124 acts as a tumor suppressor by inhibiting the expression of sphingosine kinase 1 and its downstream signaling [37]. To address this discordance, miRNA is generally divided into a 5p-arm and 3p-arm of mature miRNAs, as found in miR124-3p and miR124-5p [38], which can have opposite functions. For example, miR574-5p and miR574-3p are inversely expressed and play exactly opposite roles in gastric cancer progression [39]. They also found that miR574-5p and miR574-3p target endogenous QKI6 and ACVR1B, respectively, where miR574-5p acts as a tumor promoter and miR574-3p acts as a tumor suppressor, suggesting that 5p-arm and 3p-arm of miRNA could play a different role in tumor progression. Recently, Sueta and colleagues reported that exosomal miR124-3p was upregulated in breast cancer patients with recurrence, implying that miR124-3p promotes tumor progression [40]. Furthermore, it was revealed that miR124-3p expression was elevated in advanced triple-negative breast cancer (TNBC) patients and high miR124-3p predicts poor overall survival [41]. They also showed that miR124-3p downregulation diminished, while miR124-3p upregulation increased the growth of TNBC cells in vitro and in vivo. These findings suggest that miR124-3p, not miR124-5p, could act as an oncogene in other cancers, including HNSCC. In the present study, we found no difference in miR124-3p expression between normal tissue and chemo-radiation therapy-sensitive HNSCC, but its expression significantly increased in chemo-radiation therapy-resistant HNSCC as compared to normal tissue or sensitive HNSCC. Our study shows that miR124-3p inhibition and overexpression did not affect cell proliferation and cell motility, but miR124-3p conferred resistance to FP therapy and radiotherapy in HNSCC cells.

We found that miR124-3p could promote chemo-radiotherapy resistance by regulating the CREBRF/CREB3/ATG5 pathway. CREBRF acts as a tumor suppressor of glioblastoma through the suppression of ATG5 and CREB3 [29]. ATG5 is an autophagy family protein that is found to accelerate tumorigenesis by removing damaged molecules or organelles in lung cancer [42]. In this study, Chen and colleagues demonstrated that cisplatin treatment induced autophagy by upregulating ATG5 expression, which provides resistance to cisplatin-induced apoptosis of lung cancer cells [42]. On the other hand, CREB3 is an endoplasmic reticulum stress-related protein, which is reported to promote the malignant progression of breast cancer and glioblastoma (GBM) [43,44]. CREB3 induces GBM cell proliferation and invasion by inhibiting apoptosis via downregulating proapoptotic proteins—Bax, active caspase 3, p-PERK, p-eIF2α, and ATF4, suggesting that CREB3 might inhibit anti-cancer drug-induced apoptosis [44]. These reports support our findings that miR124-3p overexpression promotes chemo-radiotherapy resistance in HNSCC by CREBRF-mediated upregulation of ATG5 and CREB3.

MiR766-3p is another miRNA that is strongly associated with an HNSCC resistance phenotype. Overexpression of miR766-3p has been reported in other cancers, including colorectal and hepatocellular carcinoma, where it is associated with the increased proliferation and metastasis of cancer cells [32,45]. Further, it was found that miR766-3p supports cancer cell survival and metastasis by affecting Bax/Bcl-2 and EMT pathways leading to aggressive tumor behavior [46]. These data support our present findings that miR766-3p confers resistance to chemo-radiotherapy and promotes invasion and metastasis in HNSCC.

NR3C2 acts as a tumor suppressor and inhibits the proliferation, migration, and invasion of several cancers, including pancreatic, breast, lung, and glioblastoma [30,31,47,48,49]. In our study, we showed that miR766-3p decreased NR3C2 level, resulting in upregulation of c-Myc and β-catenin. It is well accepted that c-Myc and β-catenin are potent oncogenes and closely associated with drug resistance and metastasis [50,51]. Moreover, it is reported that c-Myc is upregulated in cisplatin-resistant HNSCC cell lines, and a c-Myc bromodomain inhibitor overcomes cisplatin resistance [52]. Recently, Tang and colleagues demonstrated that c-Myc promoted NRF2-driven metastasis of HNSCC via glucose-6-phosphate dehydrogenase and transketolase activation [53]. Similarly, β-catenin overexpression also led to cisplatin resistance, evasion of apoptosis, enhanced DNA repair, and enhanced migration in HNSCC cell lines [54]. Another report suggests that miR-766 targets NR3C2 and promotes hepatocellular carcinoma progression by increasing the β-catenin signaling pathway [32]. These reports strongly support our findings that miR766-3p provides resistance phenotype in HNSCC through NR3C2-mediated induction of the β-catenin/c-Myc signaling pathway. Taken together, we found that miR124-3p and miR766-3p regulate different cellular pathways regarding drug sensitivity. miR124-3p affects the CREBRF-ATG5/CREB3 axis, while miR766-3p affects the NR3C2-c-Myc/β-catenin axis; therefore, the combined targeting of these two miRNAs complement each other and shows an additive effect on sensitizing cancer cells to chemo-radiotherapy.

The limitation of this study is that we performed miRNA microarray analysis in a small number of patients. We analyzed the association between miRNA expression and survival in six resistant and six sensitive HNSCC patients. Further follow-up studies in a larger number of patients would be required to confirm our present findings and could also assess the relationship between invasiveness and miRNA expression in HNSCC patients. Future studies involving practical application in patients will be needed to definitively evaluate the clinical benefits of miRNA inhibitors.

## 5. Conclusions

Our results show that the levels of miR124-3p and miR766-3p are upregulated in non-responder HNSCC in patients. We found that expression of miR766-3p and miR124-3p is directly associated with the acquisition of a resistance phenotype, as the levels of these miRNAs increased in HNSCC cells upon acquired resistance to cisplatin and 5-FU. In vitro and in vivo studies suggested that increased expression of miR766-3p and miR124-3p supports aggressive HNSCC behavior, including drug resistance, cancer cell proliferation, and invasion. In both resistant HNSCC tumors and cell lines, the upregulation of miR766-3p and miR124-3p is associated with the downregulation of CREBRF and NR3C2—a direct target of miR124-3p and miR766-3p, respectively—which enhances tumor progression by regulating the CREB3/ATG5 and c-Myc/β-catenin signaling pathways (Figure 9C). Thus, miR124-3p-CREBRF-CREB3/ATG5 and miR766-3p-NR3C2-c-Myc/β-catenin axes play a major role in the progression and acquiring of a resistant phenotype in HNSCC. Our study implicated the potential role of miR124-3p and miR766-3p in the clinical diagnosis of HNSCC patients with resistant tumors, which might not give an appropriate response to standard chemo-radiotherapy and may indicate a requirement for alternative medical attention. Moreover, the combined targeting of miR124-3p and miR766-3p offers a powerful tool to overcome a resistance phenotype in HNSCC.

## Figures and Tables

**Figure 1 cancers-14-05273-f001:**
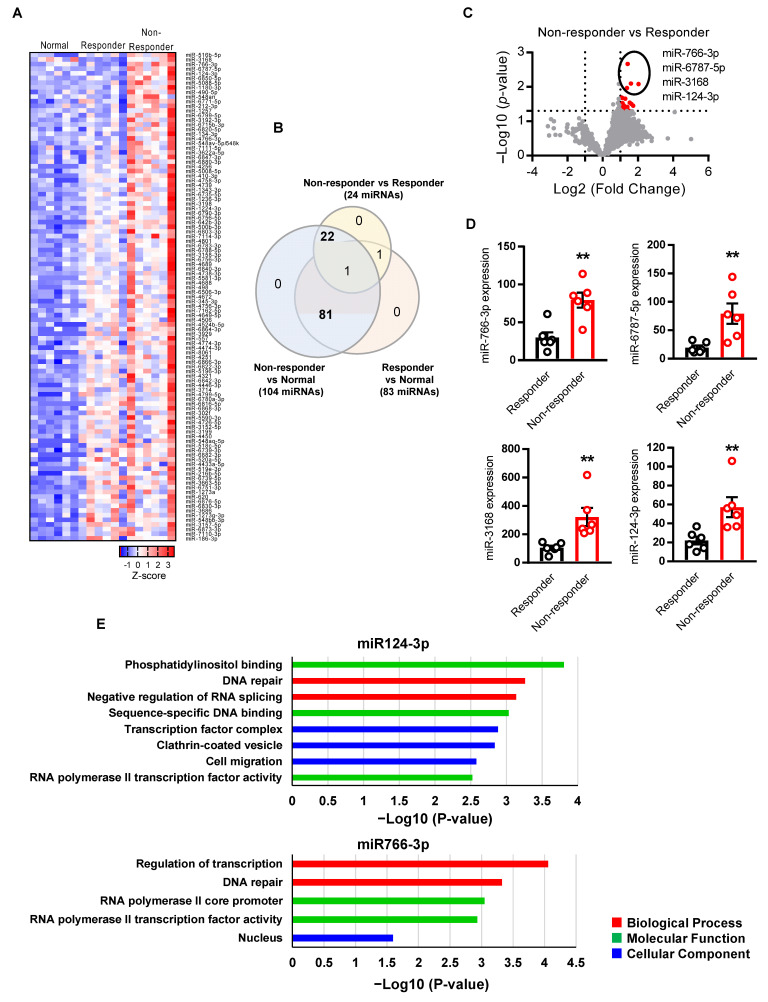
Identification of miRNA associated with HNSCC resistance. (**A**) Heatmap showing 105 significantly (*p*-value < 0.05, Fold Change > 2) differentially expressed miRNAs across three comparisons (responder vs. normal, non-responder vs. normal, and non-responder vs. responder). Each row of the heatmap represents the z-score of one differentially expressed miRNA across all samples (blue, low expression; red, high expression). (**B**) The Venn diagram showing overlaps of miRNAs that are significantly changed among three comparisons. (**C**) The volcano plots showed the miRNAs that are significantly changed in non-responder vs. responder. The red points represented upregulated miRNA. The differentially expressed miRNAs were sorted with the criteria of *p*-value < 0.05 and Fold Change > 2. (**D**) Differential expression of top four miRNA between non-responder vs. responder (n = 6 per group). (**E**) Gene ontology analysis of the miRNA target genes. The criteria of *p*-value < 0.05 was considered as significantly associated with GO pathways. A two-sided unpaired Student *t* test was used to analyze comparisons, and one-way or two-way ANOVA with Bonferroni’s correction for multiple comparisons was used to analyze group comparisons, and data are presented as means ± SEM. ** *p* < 0.01.

**Figure 2 cancers-14-05273-f002:**
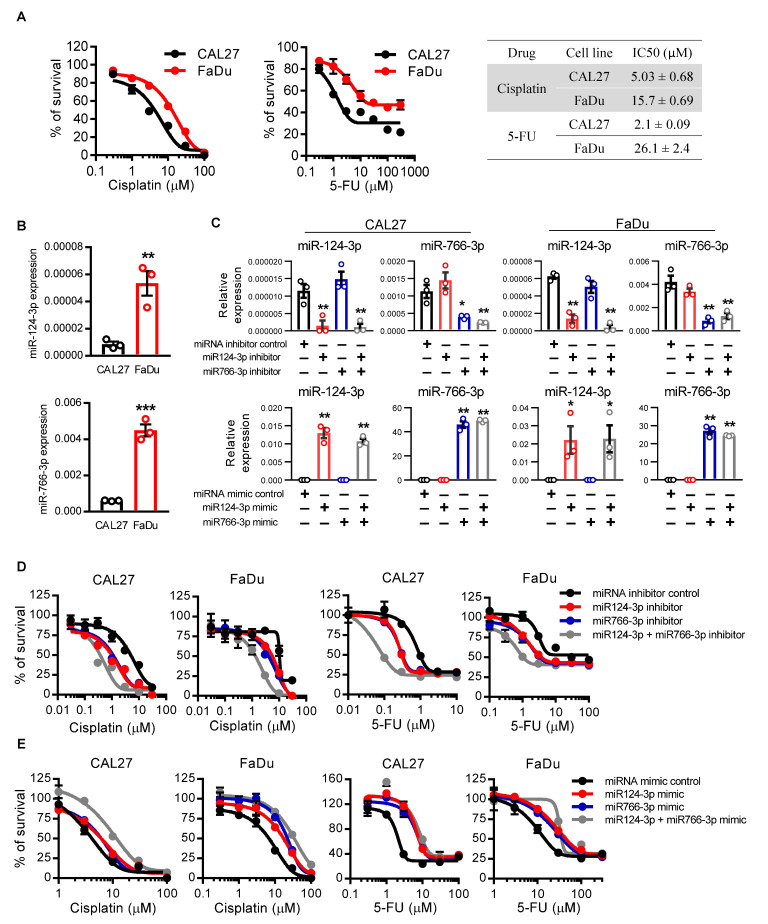
The effect of miR766-3p and miR124-3p expression on HNSCC cell sensitivity to chemotherapeutic drugs. (**A**) Dose–response curves of CAL27 and FaDu cell lines towards cisplatin and 5-FU. IC_50_ values were listed in the table. Cell viability was assessed using MTT assay. (**B**) qRT-PCR measurement of miR124-3p and miR766-3p expression in CAL27 or FaDu cell lines. (**C**) qRT-PCR measurement of miR124-3p and miR766-3p expression in CAL27 or FaDu cell lines following transfection with miRNA inhibitors (10 nM, left) or miRNA mimics (10 nM, right). (**D**,**E**) Measurement of cytotoxicity by MTT assay in CAL27 and FaDu cells in response to cisplatin or 5-FU ± 24 h prior transfection with 10 nM miRNA inhibitors (**D**) or miRNA mimics (**E**). IC_50_ values are listed in the Appendix A. Two-sided unpaired Student t test was used to analyze comparisons, and one-way ANOVA with Bonferroni’s correction for multiple comparisons was used to analyze group comparisons. Data are presented as means ± SD (n = 3). Each experiment was performed three times, and each time it included triplicate samples. * *p* < 0.05, ** *p* < 0.01 and *** *p* < 0.001.

**Figure 3 cancers-14-05273-f003:**
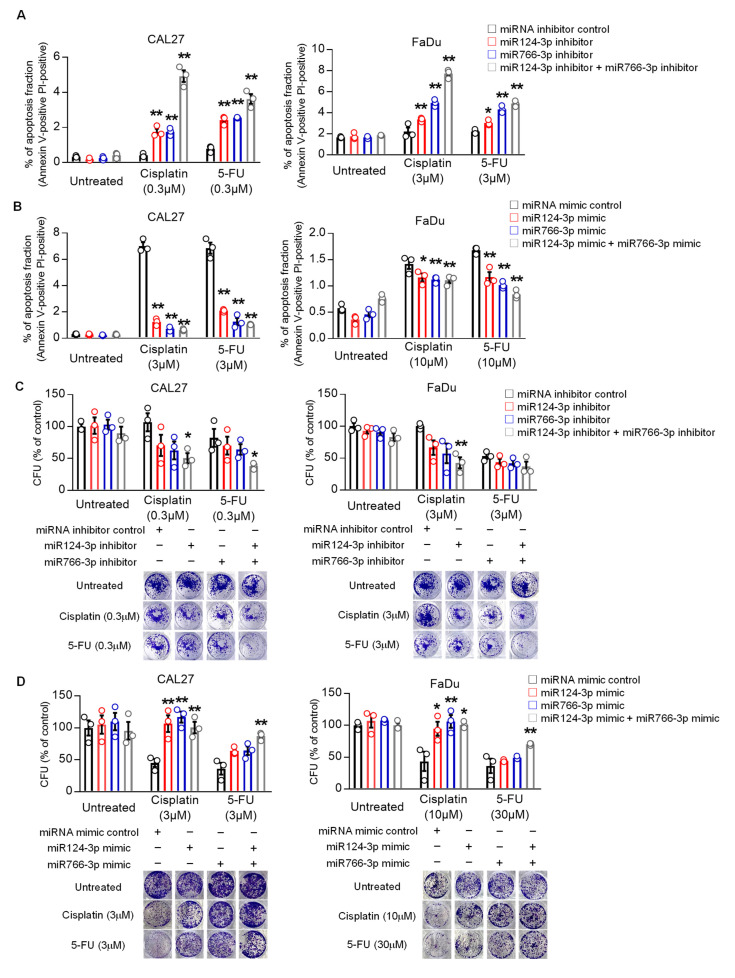
The effect of miR766-3p and miR124-3p expression on HNSCC cell apoptosis and clonogenic potential. (**A**,**B**) Measurement of apoptosis in CAL27 or FaDu cells in response to cisplatin or 5-FU ± 24 h prior to transfected with 10 nM miRNA inhibitors (**A**) or miRNA mimics (**B**). After 24 h of drug treatments, cells were stained for annexin V (FITC) and propidium iodide (PI) and then analyzed with flow cytometry. Dot plots data from this analysis are presented in Appendix A. (**C**,**D**) Measurement of clonogenic potential by colony formation assay. Cancer cells were treated with either 10 nM miRNA inhibitors (**C**) or miRNA mimics (**D**) for 24 h. Then, medium (DMEM + 10%FBS) was changed with or without indicated drug. Cells were allowed to grow for additional seven days, and after staining by crystal violet, colony formation units (CFU) were counted. A one-way or two-way ANOVA with Bonferroni’s correction for multiple comparisons was used to analyze group comparisons, and data are presented as means ± SD (n = 3). Each experiment was performed three times, and each time it included triplicate samples. * *p* < 0.05 and ** *p* < 0.01.

**Figure 4 cancers-14-05273-f004:**
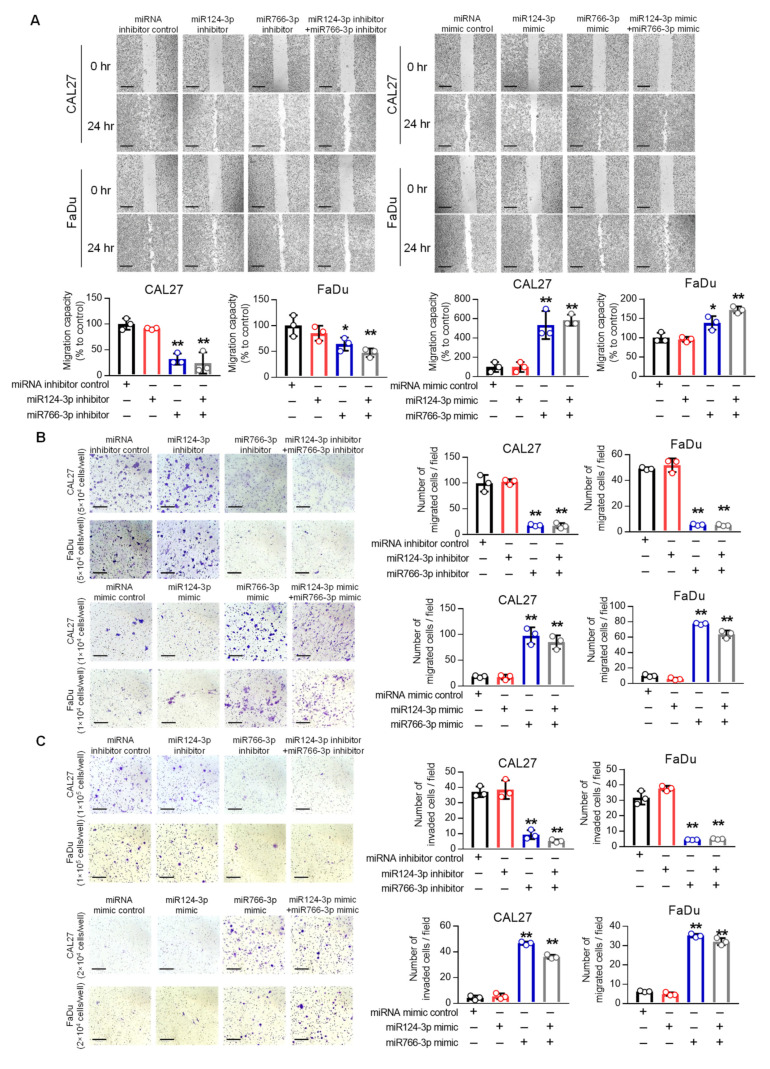
The effect of miR766-3p and miR124-3p expression on HNSCC cell motility. (**A**) Measurement of cell migration by scratch wound healing assay in CAL27 and FaDu cells following transfection with 10 nM miRNA inhibitors (left) or miRNA mimics (right). The images were taken at 0 and 24 h after scratching the confluent cell monolayer with micropipette tip (scale bar, 520 µm). The length of the gap was measured for each sample and presented as the average ratio of residual gap to the initial gap in graphs. (**B**,**C**) Determination of cell migration and invasion by transwell assay. Following transfection with either miRNA inhibitors (top) or mimics (bottom), cells were seeded on the upper chamber of a transwell insert, and cell migration to the lower chamber (**B**) or invasion in Matrigel (**C**) was determined as described in methods. The migrated or invaded cells are presented as average number per field in triplicate (Scale bar, 210 µm). A one-way ANOVA with Bonferroni’s correction for multiple comparisons was used to analyze group comparisons, and data are presented as means ± SD (n = 3). * *p* < 0.05 and ** *p* < 0.01.

**Figure 5 cancers-14-05273-f005:**
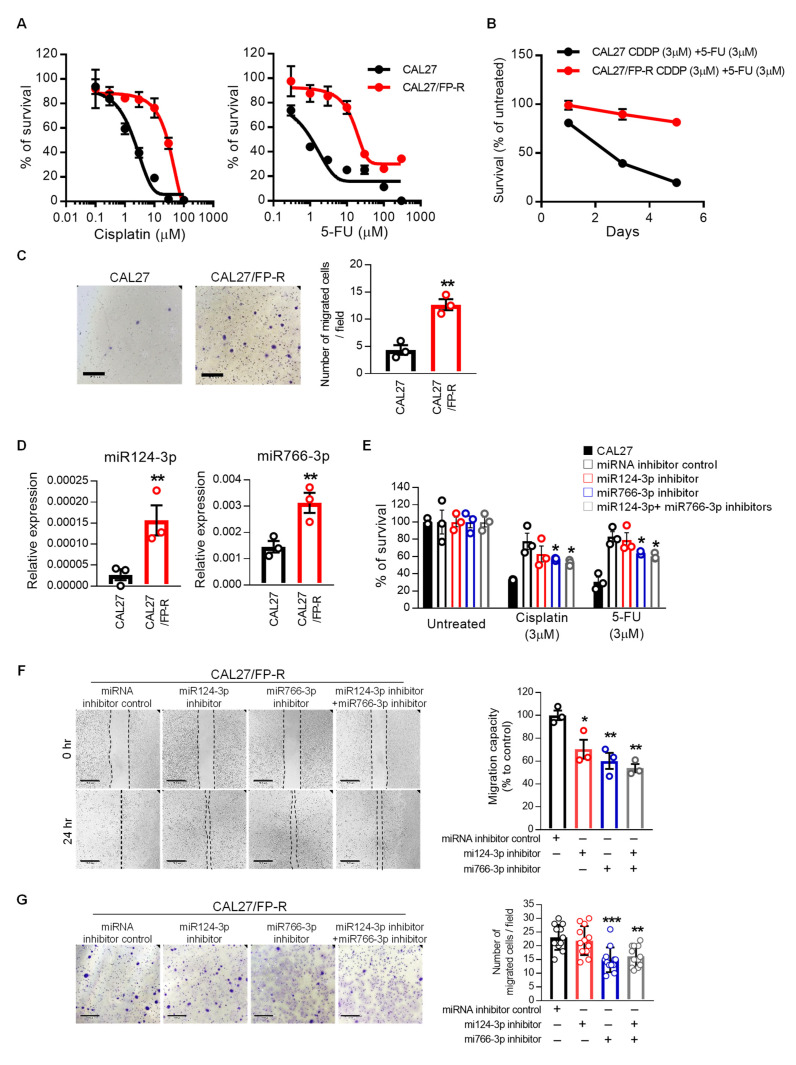
The upregulation of miR124-3p and miR766-3p is associated with acquired resistance in HNSCC cells. (**A**) Dose–response curves of the resistant CAL27/FP-R cell line towards cisplatin and 5-FU. Cell viability was assessed using MTT assay. (**B**) The effect of cisplatin and 5-FU on cell survival in CAL27 and CAL27/FP-R cells. Cell survival was assessed using an MTT assay on Days 1, 3, and 5 after treatment with cisplatin and 5-FU combination. (**C**) Determination of cell migration by Transwell chamber-migration assay in CAL27/FP-R cells, as described in Figure 4. (**D**) Relative level of miR124-3p and miR766-3p in the CAL27/FP-R cell line, measured by qRT-PCR. (**E**) The effect of miRNA inhibition on drug-induced cytotoxicity in the resistant cell line (CAL27/FP-R). Cells were transfected by 10 nM inhibitor (single or combined) for 24 h followed by 72 h exposure to the indicated drug. Cytotoxicity was determined by MTT assay. * *p* < 0.05 (vs. miRNA inhibitor control). (**F**) Measurement of cell migration by scratch wound healing assay following 24 h treatment with 10 nM miRNA inhibitor (single or combined). (**G**) Measurement of cell migration by Transwell chamber-migration assay following 24 h treatment with 10 nM miRNA inhibitor (single or combined). Each dot represents 1 field × 4 images per transwell × triplicate = total 12 fields per treatment condition. These assays are described in Figure 4. A one-way ANOVA with Bonferroni’s correction for multiple comparisons was used to analyze group comparisons, and data are presented as means ± SD (n = 3). * *p* < 0.05, ** *p* < 0.01 and *** *p* < 0.001.

**Figure 6 cancers-14-05273-f006:**
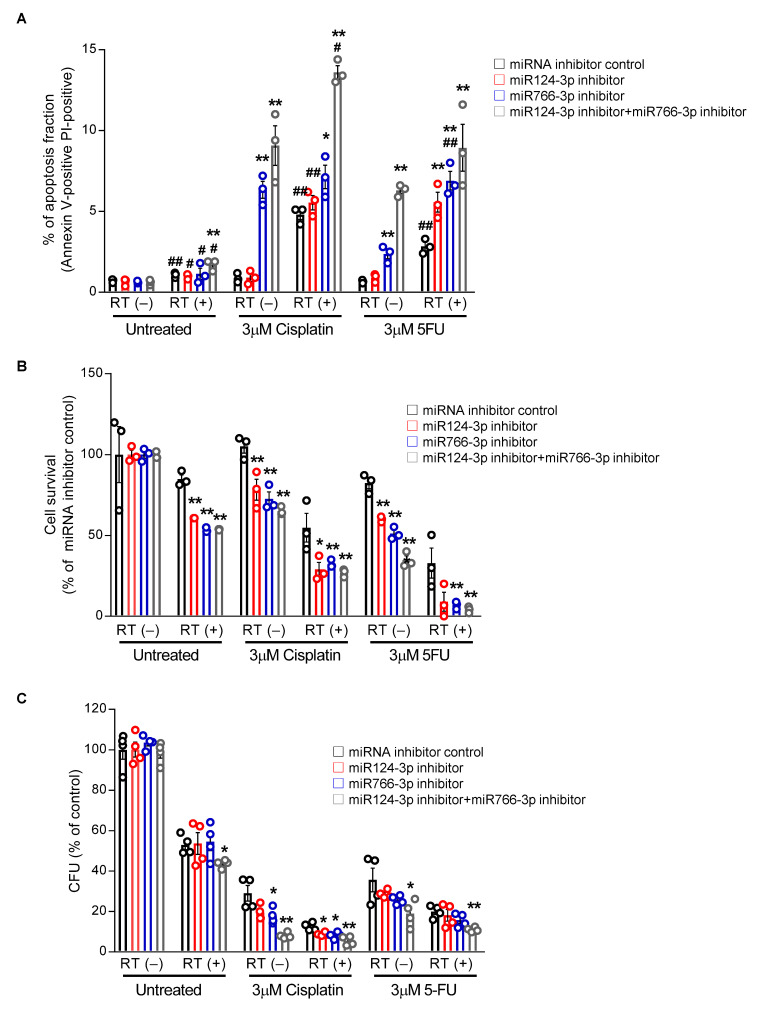
The effect of miR124-3p and miR766-3p inhibition on sensitizing resistant HNSCC cells to chemo-radiation therapy. (**A**) Measurement of apoptosis. Following transfection with miRNA inhibitors (10 nM each) for 24 h, CAL27/FP-R cells were treated with radiotherapy (8 Gy) alone or in combination with chemotherapeutic drug (cisplatin or 5-FU). After 24 h, cells were labeled with anti-annexin V-FITC antibody and PI and analyzed by flow cytometry. Dot plot data are shown in Appendix A. (**B**) Measurement of drug-induced cytotoxicity. 24 h after transfection with miRNA inhibitors (10 nM each), CAL27/FP-R cells were exposed to 8 Gy radiation, and then with or without chemotherapy treatment (cisplatin or 5-FU) for 72 h, cytotoxicity was assessed by MTT assay. (**C**) Determination of clonogenic potential. CAL27/FP-R cells were transfected for 24 h and then exposed to 8 Gy radiation. With or without chemotherapeutic drug treatments as indicated, cells were stained with crystal violet, and CFUs were counted. A two-way ANOVA with Bonferroni’s correction for multiple comparisons was used to analyze group comparisons, and data are presented as means ± SD (n = 3). * *p* < 0.05, ** *p* < 0.01 (vs. miRNA inhibitor control). [# *p* < 0.05, ## *p* < 0.01 (vs. RT(-)].

**Figure 7 cancers-14-05273-f007:**
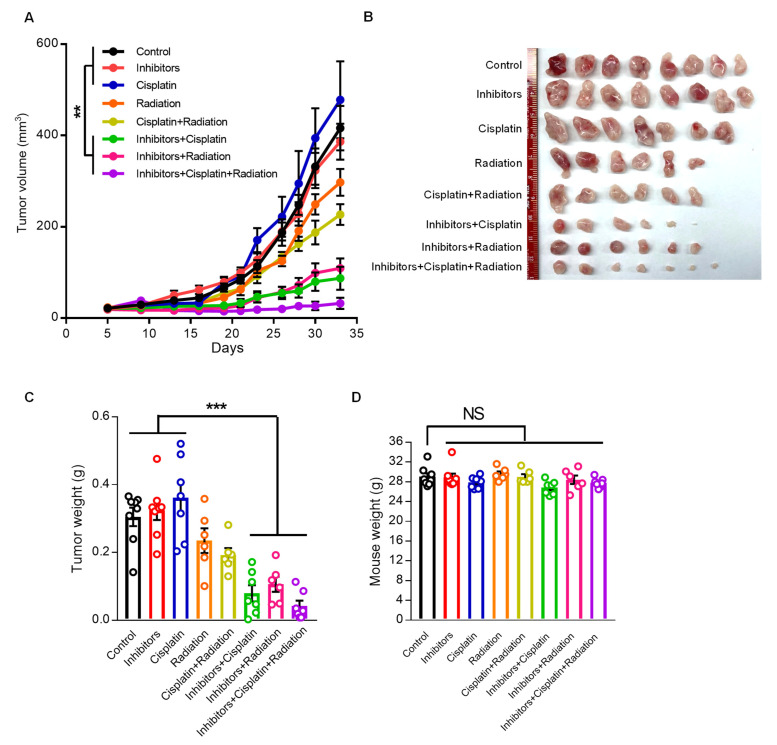
The effect of miR124-3p and miR766-3p inhibition on resistant HNSCC progression. (**A**–**D**) Anti-cancer effect of cisplatin (2.5 mg/kg/mouse, i.p., every three days), miRNA inhibitors (1 µg/tumor, every three days, intratumor injection), or radiation (8 Gy/tumor, Day 7) as monotherapy and multimodality therapy regimens on xenograft mouse model bearing CAL27/FP-R resistant tumors. CAL27/FP-R cells were implanted into the subcutaneous tissue of the right abdominal wall of male nude mice. After tumor was established, mice were sorted into eight groups and treated with either cisplatin or miRNA inhibitors or radiation alone or in combination as cisplatin+radiation or miRNA inhibitors+cisplatin or miRNA inhibitors+radiaition or miRNA inhibitors+cisplatin+radiaition. The plot depicting comparison of tumor growth rates (**A**). The representative images (**B**) and weights (**C**) of CAL27/FP-R tumors are shown. (**D**) Showing mice weights at day 33. A one-way ANOVA with Bonferroni’s correction for multiple comparisons was used to analyze group comparisons, and data are presented as means ± SEM (n = 6–8 per group). ** *p* < 0.01 and *** *p* < 0.001.

**Figure 8 cancers-14-05273-f008:**
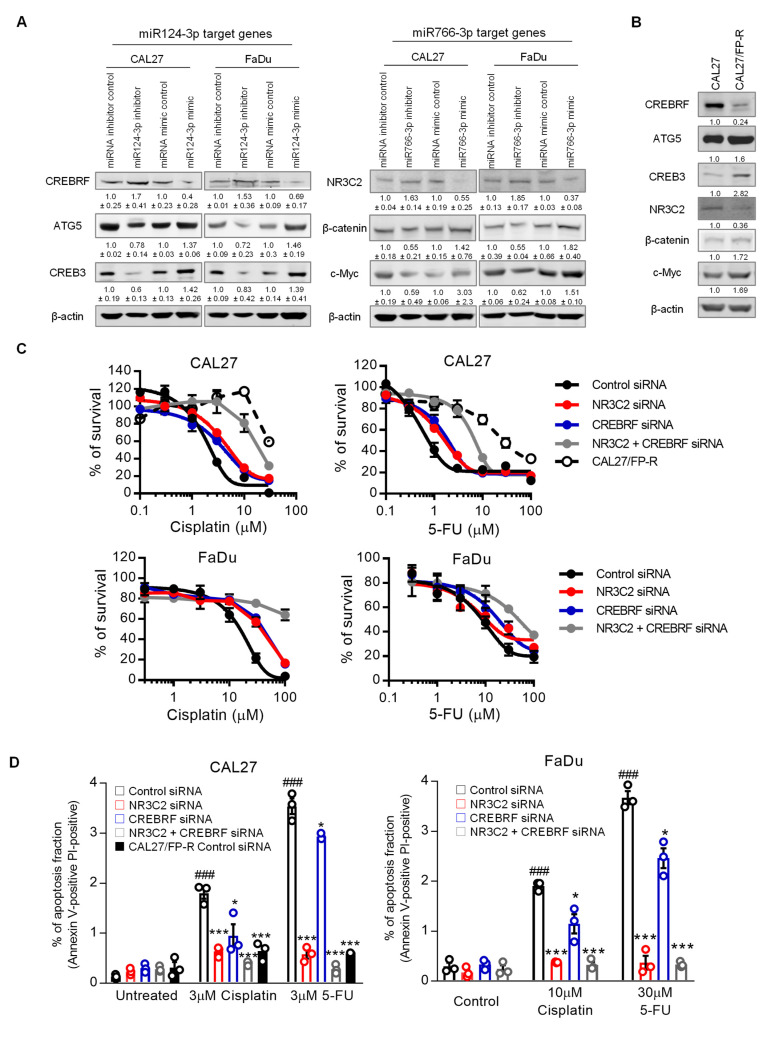
The role of miR124-3p and miR766-3p target genes in HNSCC drug resistance. (**A**) Expression analysis of miR124-3p and miR766-3p direct target genes and downstream target genes by Western blot in HNSCC cell lines (CAL27 and FaDu), with or without transfection with miRNA inhibitors or miRNA mimics. Left: Western blotting showed the expression of miR124-3p target gene (CREBRF) and CREBRF target genes (ATG5 and CREB3). Right: Western blotting showed the expression of miR766-3p target gene (NR3C2) and NR3C2 target genes (β-catenin and c-Myc). Quantitative data (relative expression levels after β-actin-corrected) from three independent experiments are disclosed below each protein band. Data represent the mean ± SD (n = 3). (**B**) Target gene analysis in sensitive (CAL27) vs. resistant (CAL27/FP-R) HNSCC cell lines. Quantitative data (relative expression levels after β-actin-corrected) is shown below each protein band. (**C**) The effect of NR3C2 and/or CREBRF knockdown on drug-induced cytotoxicity in CAL27 and FaDu. Cells were transfected by 10 nM siRNA (single or combined) for 24 h followed by 72 h exposure to the indicated drug. Cytotoxicity was determined by MTT assay. IC_50_ values are listed in Appendix A. (**D**) Measurement of apoptosis in CAL27 or FaDu cells in response to cisplatin or 5-FU ± 24 h prior transfection with 10 nM siRNA. After 24 h of drug treatments, cells were labeled with anti-annexin V-FITC antibody and PI and then analyzed with flow cytometry. A two-way ANOVA with Bonferroni’s correction for multiple comparisons was used to analyze group comparisons, and data are presented as means ± SD (n = 3). * *p* < 0.05, *** *p* < 0.001 (vs. control siRNA), and ### *p* < 0.001 (vs. untreated).

**Figure 9 cancers-14-05273-f009:**
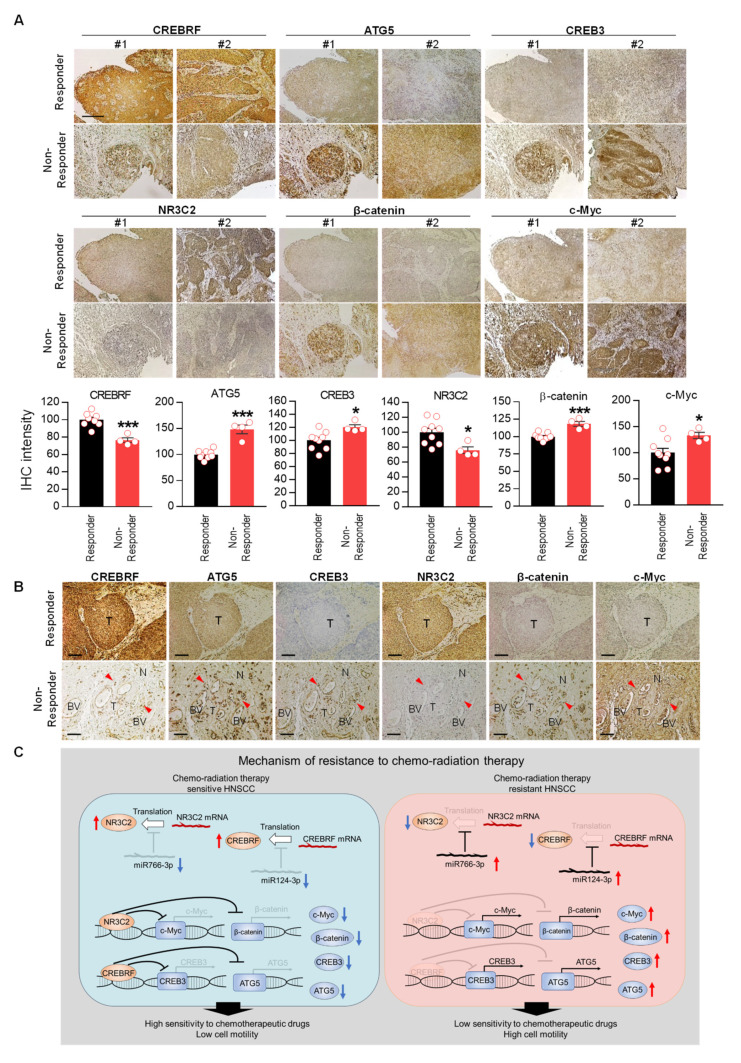
Downregulation of CREBRF and NR3C2 increase poor prognosis in HNSCC. (**A**) Histological analysis of miR124-3p and miR766-3p target gene expression (CREBRF-ATG5/CREB3, NR3C2-β-catenin/c-Myc) in Responder vs. Non-Responder HNSCC clinical samples. Magnification, x100. Scale bar, 210 µm. The quantitative data from all specimens are shown in the bar chart. Each dot in the graph represents an individual clinical sample. Two-sided unpaired Student t test was used to analyze comparisons, and data are presented as means ± SEM. * *p* < 0.05 and *** *p* < 0.001. (**B**) Histological analysis of tumor morphology in relation to miR124-3p and miR766-3p target gene expression. Representative images of CREBRF, ATG5, CREB3, NR3C2, β-catenin, and c-Myc expression in the serial section of responder and non-responder HNSCC specimens. BV: Blood Vessel. T: Tumor. N: Normal tissue. The invasive cancer cells are indicated by red arrowhead. Magnification, ×200. Scale bar, 100 µm. (**C**) Summary of resistance mechanisms regulated by miR124-3p and miR766-3p. Our data indicated that upon acquired resistance in HNSCC cells or in non-responder HNSCC tumors, the levels of miR124-3p and miR766-3p go up, which in turn down-regulate its direct target genes: CREBRF and NR3C2, and consequently the expression of downstream targets of CREBRF (ATG5/CREB3) and NR3C2 (β-catenin/c-Myc) increased in resistant tumors, which are positively correlated with poor prognosis. Thus, by enhancing the CREBRF-ATG5/CREB3 and NR3C2-β-catenin/c-Myc axis, miR124-3p and miR766-3p support aggressive HNSCC progression.

**Table 1 cancers-14-05273-t001:** Clinical characteristics of HNSCC patients.

Tissue Type	Age at Diagnosis	Gender	Location	Histology	Clinical Stage	Treatment(s)	PFS or OS
Responder #1	47	Male	Left supraglottic mass	Invasive squamous cell carcinoma, Poorly differentiated, focally keratinizing	T4	Primary Tonsillar cancer: Surgery 1st Recurrence: Radiation 2nd Recurrence: Cisplatin+Taxol	PFS 1st recurrence: 60 months PFS 2nd recurrence: 156 months OS: 300 months
* Responder #2	68	Female	Tongue	Invasive squamous cell carcinoma, Non-keratinizing type	cT2N0	Chemo-RT (Cisplatin)	OS: 81 months
* Responder #3	57	Male	Tonsil	Basaloid squamous cell carcinoma	pT2N0	Induction Chemo-RT (Cisplatin) Recurrence: cisplatin + radiation	PFS: 20 months OS: 109 months
* Responder #4	53	Male	Left maxilla	Invasive squamous cell carcinoma, Keratinizing moderately differentiated	pT4aN0M0	Chemo-RT (Cisplatin)	OS: 84 months
Responder #5	73	Male	Tongue base	Squamous cell carcinoma, Non-keratinizing	T3N1	Chemo-RT (Cisplatin)	OS: 80 months
* Responder #6	54	Male	Left orbit/nasal	Invasive squamous cell carcinoma, Moderately–poorly differentiated	pT4aN0	Chemo-RT (Cisplatin)	OS: 55 months
* Non-Responder #1	57	Male	Tonsil	Invasive squamous cell carcinoma, Poorly differentiated	T2N0	Adjuvant; Chemo-RT (Carboplatin) Disease progression: (Cetuximab)	PFS: 1 month OS: 4 months
Non-Responder #2	76	Male	Tongue	Invasive squamous cell carcinoma, Keratinizing moderately differentiated	T4aN2M0	Chemo-RT (Carboplatin+Taxol)	PFS: 6 months OS: 16 months
* Non-Responder #3	68	Female	Left septum	Invasive squamous cell carcinoma, Keratinizing moderately-poorly differentiated	pT1	Chemo-RT (Carboplatin)	PFS: 7 months OS: 30 months
Non-Responder #4	76	Male	Tongue	Invasive squamous cell carcinoma, acantholytic, Keratinizing poorly differentiated	T2N1	Chemo-RT (Carboplatin)	PFS: 5 months OS: 17 months
Non-Responder #5	55	Male	Tonsil	Invasive squamous cell carcinoma, Poorly differentiated	T1N2bM0	Adjuvant Radiation Recurrence: Chemo (Cisplatin + 5FU)	PFS: 13 months OS: 28 months
Non-Responder #6	68	Female	Nasopharynx	Poorly differentiated squamous cell carcinoma	T4N2	Chemo (Cisplatin + 5FU)	OS: 5 months

RT: Radiation Therapy; PFS: Progression-free survival; OS: Overall survival. Note: Normal tissues were collected from the adjoining area of tumors on patients marked with *.

## Data Availability

All data generated or analyzed during this study are included either in this article or in the Appendix A.

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
