# Peer review of "miR766-3p and miR124-3p Dictate Drug Resistance and Clinical Outcome in HNSCC"

_cancers, 2022, doi:10.3390/cancers14215273_

Round 1

Reviewer 1 Report

I found this paper interesting and results clinically valuable.

I only suggest authors to briefly discuss limitations of the paper and predict targets for studied miRNAs.

Author Response

Comments: I found this paper interesting and results clinically valuable. I only suggest authors to briefly discuss limitations of the paper and predict targets for studied miRNAs.

Response: Thank you for your comments. We have incorporated the changes suggested by the reviewer in the revised MS.

Thank you again for your time to review our paper. I hope with the addition of new information, the revised manuscript is suitable for publication.

Reviewer 2 Report

This manuscript is very interesting, authors investigated role of miR766-3p and miR124-3p in drug resistance in HNSCC. I commend authors for strong work. This manuscript is well written, it addresses important questions and would be help towards developing new therapies for HNSCC.

A minor comment: Western blots for ATG5, CREB3 and CREBRF has lot of background bands. I believe this is because of too much of secondary antibody. Secondary antibody reduction would help in clear blots.  

Author Response

Comments: This manuscript is very interesting, authors investigated role of miR766-3p and miR124-3p in drug resistance in HNSCC. I commend authors for strong work. This manuscript is well written, it addresses important questions and would be help towards developing new therapies for HNSCC.

A minor comment: Western blots for ATG5, CREB3 and CREBRF has lot of background bands. I believe this is because of too much of secondary antibody. Secondary antibody reduction would help in clear blots.  

Response: Thank you for your comments. As suggested, the expression of ATG5, CREB3 and CREBRF has been reanalyzed and revised Western blots are presented in Figure 8 and Figure S7. 

Thank you again for your time to review our paper. I hope with the new incorporated information, the revised manuscript is acceptable for publication.

Reviewer 3 Report

The paper is well written and provides a good introduction to the topic of oncologic therapy resistance. some comments i would like to make. Please dicuss the primary pillar of oncologic therapy, which is surgical tumor removal. In Material and Methods, conventional therapy is mentioned. Did the subjects have surgical tumor resection, neck dissection and or reconstruction? How many were Ro- resections, what was the lymph node status? Please also discuss the limitations of the study?

I would like to express my appreciation to the authors for their work.

Author Response

Comments: The paper is well written and provides a good introduction to the topic of oncologic therapy resistance. some comments I would like to make. Please discuss the primary pillar of oncologic therapy, which is surgical tumor removal. In Material and Methods, conventional therapy is mentioned. Did the subjects have surgical tumor resection, neck dissection and or reconstruction? How many were Ro- resections, what was the lymph node status? Please also discuss the limitations of the study. I would like to express my appreciation to the authors for their work.

Response: We agree with the reviewer’s view point that surgical removal is one of the main strategies for treating cancer, if possible. We have discussed this in the paper. However, in the advanced stages or for recurrent tumors, adjuvant therapies are primary line of treatments. Our paper is limited to examining the mechanism of HNSCC drug resistance mediated by miRNA, particularly to FP chemotherapy and radiotherapy.   

The clinical characteristics of HNSCC patients are shown in Table 1. As suggested by the reviewer, we have elaborated this information in the methods (lines 104 – 110). Specifically, at the initial diagnosis, the clinical stage ranges from T2 to T4 and T1 to T4a in responder and non-responder group, respectively. For both groups, nodal status ranges from N0-N2b and no metastasis was identified. The initial procedure of all samples included biopsy (10/16), local resection (5/16) and neck dissection (1/16). Two patients in the responder group had metastatic disease rendered from re-resection, but without lymph node metastasis.

The limitations have also been discussed in the revised paper.

Thank you again for your time to review our paper. I hope with these changes and newly incorporated information, the revised manuscript is acceptable for publication.